# Cosmological infrared subtractions & infrared-safe computables

**Paolo Benincasa[1,2]⋆ and Francisco Vazão[1]†**

**1** Max-Planck-Institut für Physik, Werner-Heisenberg-Institut,
Boltzmannstrasse 8, D-85748 Garching, Germany
**2** Instituto Galego de Física de Altas Enerxías IGFAE,
Universidade de Santiago de Compostela, E-15782 Galicia-Spain

⋆ pablowellinhouse@anche.no , † fvvazao@mpp.mpg.de

## Abstract

Cosmological observables in perturbation theory turn out to be plagued with infrared divergences, which represents both a conceptual and computational challenge. In this paper we present a proof of concept for a systematic procedure to remove these divergences in a large class of scalar cosmological integrals and consistently define an infrared safe *computable* in perturbation theory. We provide diagrammatic rules which are based on the nestohedra underlying the asymptotic structure of such integrals.



# 1 Introduction

One of the deepest challenges in fundamental physics is the understanding of the mechanism in the early Universe which set up the initial conditions for the evolution that lead to the cosmic structures we can observe. Inflation provides the current leading proposal: The Universe underwent a phase of accelerated expansion that stretched quantum fluctuations to cosmic scales, seeding the formation of the large scale structures. These seeds are nothing but equal-time correlation functions at the end of inflation, which are the end point of the early-time inflationary evolution.

When sufficiently light states are involved, equal-time correlation functions in an expanding background are known to show infrared divergences [1–38] that have been argued to possibly make the cosmological space-time unstable [1–6, 8, 21]. These divergences are typically of two types, one is associated to the loop momentum integration when massless fields are exchanged. The other is usually referred to as a secular divergence, and is associated to the accumulation of long wavelength modes in at late times (due to the expanding background in which the modes propagate).

However, the analysis via different approaches – carried out mainly for massless scalars in a (1+3)-dimensional de Sitter background – suggests that the leading infrared effects resum [14, 23, 28, 30], with the resummation of the low-energy modes providing a probability distribution that satisfies a Fokker-Planck equation [11, 28, 30, 37, 39–44] in agreement with Starobinski's stochastic inflation [45, 46]. Despite the approaches developed in [44] and [28] are suited for a more general analysis that should allow for a complete understanding of the subleading divergences, much less is known about them, as it is much less known the fate of the infrared divergences in arbitrary FRW backgrounds. More generally, given an equal-time observable in an expanding background with infrared divergences:

❏ What are the conditions it ought to satisfy for the divergences to resum?

❏ Is it possible to have a first principle definition of an infrared finite observable?

In this paper, we provide a proof of concept about the possibility of addressing these questions. We build upon the recent progress in the understanding the analytic structure of cosmological observables, in particular the Bunch-Davies wavefunction and the in-in correlators – see [47–53] and references therein – including their asymptotic structure [54].

The Bunch-Davies condition implies that cosmological observables in perturbation theory can show singularities on the loci identified by the vanishing of positive linear combinations of the energies,[1] and these loci are located outside the physical region as the energies are all non-negative.[2] When certain light, as well as conformally coupled, scalars are involved, these perturbative observables can be expressed in terms of integrals of a universal integrand, constituted by the relevant flat-space observable, over its energies, with the measure of integration encoding the cosmology as well as the states involved [55, 56]. In this language, the infrared effects due to the expansion of the Universe are encoded for large energies [54], while the infrared region due to loop mode is related to soft momenta associated to each of the loop edges.[3] For Feynman graphs contributing to the perturbative wavefunction, the soft loop region does not give rise to any singularity and consequently the infrared singularity, when present, are due to the expansion of the Universe only. This is in contrast to what happens for in-in correlation functions, whose infrared divergent structure emerges from both these effects.

---

[1]With abuse of language, the word *energy* is used to identify the modulus of (the sum of) momenta.

[2]In the physical region, these loci can be reached just whether the energies vanish, *i.e.* for trivial kinematics.

[3]In the physical region, there is no loop infrared effect due to collinear limits, as these sheet lie outside it.

These integrals posses the remarkable feature that their integrands are encoded in the combinatorics of cosmological polytopes[4] [48, 53, 55–57]. Their face structure encode the singularity structure of the cosmological observable, with the facets (*i.e.* the codimension-1 faces) related to the coefficients of the codimension-1 singularities [55, 58, 59], while the compatibility conditions among facets provide the compatibility of sequential discontinuities in codimension-2 [60] and higher [61]. It also directly determines the asymptotics of the integral, as it is responsible for the peculiar structure of the associated Newton polytope [54] which codifies the convergence region of the integral [62, 63] and allows isolating the divergences through sector decomposition [64–70]. For the class of integrals we are interested in, the Newton polytopes turn out to be nestohedra [54], *i.e.* polytopes that can be constructed as a Minkowski sum of simplices of different codimension [71, 72], allowing for a simpler determination of the compatibility conditions among its facets and, consequently, of the sectors contributing to a given divergence. As we will review, the facets of a given nestohedron are determined by a covector which encodes both the direction in which the integral becomes divergent, and its degree of divergence. Furthermore, such a covector is associated to tubings of the graph associated to the underlying cosmological polytope. The combinatorics of the nestohedra, together with its realisation in terms of tubings of a graph, allows for a more straightforward implementation of sector decomposition and extract both leading and subleading divergences [54].

In this paper, we use this combinatorial analysis to formulate a systematic, diagrammatical, procedure for subtracting the infrared divergences. It is based on the idea of identifying those graph whose associated observable share the same asymptotic behaviour. This can be translated into graphical rules, for which the original graph has to be subtracted by all the graphs that can be obtained from it by collapsing two adjacent sites – a graph obtained in this way has one site and one edge less. This allows to construct a infrared finite *computable* for an arbitrary graph.

A subtraction procedure based on holographic renormalisation has been recently proposed in [38], where the discussion is framed in terms of the bulk fields in $(1 + d)$-dimensional de Sitter space. Our perspective is somewhat different as our subtraction procedure does not aim to renormalise the infrared divergences, rather it consistently defines a perturbative infrared-safe computable based on the analytic structure in the infrared region in a similar (but not quite) spirit as done for flat-space Feynman integrals [73]. It is important to stress that our construction on one side is order by order in perturbation theory, and on the other constitutes *a proof of concept* about the possibility of defining such quantities. For example, in the flat-space case, it has been known for long time that a certain combination at one loop between the box diagram and the triangle ones is infrared finite [73], there exist an observable, the Wilson loops with a Lagrangian insertion [74–76], whose perturbative expansion [77] returns precisely this combination and has a geometrical interpretation [78]. In our case, a more first principle definition of a computable whose perturbative expansion returns our infrared-safe diagrams (or a modification of them) is needed as well as a combinatorial first principle definition along similar lines of the cosmological polytopes. We leave this to future work.

In what follows, we begin with providing a general description of the cosmological integrals of interest, their relation to graphs and to cosmological polytopes – here we provide an essential review of what is needed in the present paper, and the interested reader is referred to the original literature or the reviews [48, 49]. We also review the asymptotic analysis for these integrals, discussing how the infrared behaviour is encoded into the nestohedra and how

---

[4]With a little abuse of language as well as for brevity, we refer as *cosmological polytope* to any of the combinatorial structures that arose in the study of the perturbative structure of cosmological processes, *i.e.* the originally formulated cosmological polytopes describing the Bunch-Davies wavefunction for conformally coupled scalars [55], the generalised cosmological polytopes for the wavefunction with light states [56,57], and the weighted cosmological polytopes for in-in correlators [53].

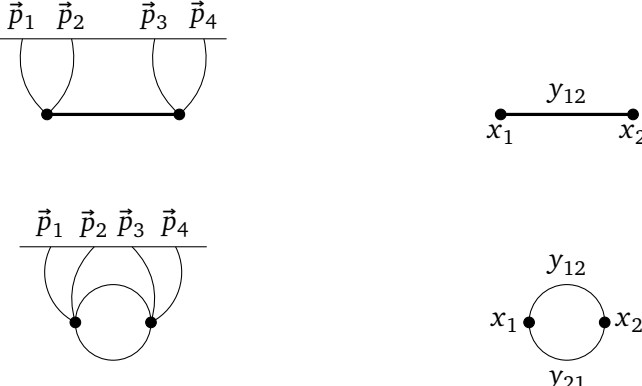

Figure 1: From Feynman diagrams to weighted reduced graphs. Feynman diagrams (*left column*) show lines each of which runs from a site to the boundary (the upper horizontal line), representing the external states, and edges connecting the sites, representing the bulk-to-bulk propagation. They can be mapped into weighted reduced graphs (*right column*) by erasing the external state lines and the boundary and assigning weights $\{x_s\ s \in \mathcal{V}\}$ to the sites and $\{y_e, e \in \mathcal{E}\}$ to the edges.

it can be extracted. This allows us to formulate our subtraction procedure and define an infrared finite quantity for each graph. We separately discuss tree and loop graphs, and, for each of these two cases, the logarithmic and power-law divergences. While the main text contains the general discussion, explicit examples are reported in the appendix.

## 2 Cosmological integrals

In this paper, we consider the class of integrals emerging from the description of the perturbative wavefunction and correlators in terms of cosmological polytopes[5] [53, 55, 56, 60] – both observables are taken to satisfy the Bunch-Davies condition in the infinite past. Such a description accounts for certain scalar light states in FRW cosmologies with polynomial interactions.

In a nutshell, let us consider a Feynman diagram contribution to the perturbative wavefunction. It can be mapped into a *weighted reduced graph* by suppressing the external states and assigning the weights $\{x_s, s \in \mathcal{V}\}$ and $\{y_e, e \in \mathcal{E}\}$ to its sites and edges respectively – see Figure 1. The sets $\mathcal{V}$ and $\mathcal{E}$ are respectively the set of sites and edges of $\mathcal{G}$, with $n_s = \dim\{\mathcal{V}\}$ and $n_e = \dim\{\mathcal{E}\}$.

Given a weighted reduced graph $\mathcal{G}$, a subgraph $\mathfrak{g} \subseteq \mathcal{G}$ is associated to a linear polynomial $q_{\mathfrak{g}}(\mathcal{Y})$

$$q_{\mathfrak{g}}(\mathcal{Y}) = \sum_{s \in \mathcal{V}_{\mathfrak{g}}} x_s + \sum_{e \in \mathcal{E}_{\mathfrak{g}}^{\text{ext}}} y_e \,, \tag{1}$$

where $\mathcal{Y} := (\{x_s\}_{s \in \mathcal{V}}, \{y_e\}_{e \in \mathcal{E}})$ is a vector formed by all the weights of the graph, $\mathcal{V}_{\mathfrak{g}} \subseteq \mathcal{V}$ is the subset of sites in $\mathfrak{g} \subseteq \mathcal{V}$, $\mathcal{E}_{\mathfrak{g}}^{\text{ext}}$ is the subset of edges departing from $\mathfrak{g}$. In words, the linear polynomial is given by the sum of the weights associated to the sites in $\mathfrak{g}$ and the edges departing from it. Each of the vanishing loci $\{q_{\mathfrak{g}}(\mathcal{Y}) = 0, \mathfrak{g} \subseteq \mathcal{G}\}$ identifies a singularity of the associated wavefunction. Such singular points live outside the physical region, as the Bunch-Davies condition in the infinite past requires that each weight is non-negative and hence these

---

[5]For brevity, we will indicate with *cosmological polytopes* all the combinatorial structures –namely, the cosmological polytopes themselves [55], the generalised cosmological polytopes [56, 60] and the weighted cosmological polytopes [53] – that arose in the studies of cosmological processes, unless otherwise specified.

vanishing loci in the physical region would correspond to trivial kinematics. Consequently, in the physical region $q_{\mathfrak{g}}(\mathcal{Y}) > 0$, $\forall \mathfrak{g} \subseteq \mathcal{G}$. Thinking of $\{q_{\mathfrak{g}}(\mathcal{Y}) = 0, \mathfrak{g} \subseteq \mathcal{G}\}$ as a set of hyperplanes in $\mathbb{P}^{n_s+n_e-1}$, with the weights forming a set of local coordinates and $\mathcal{Y}$ being a generic point in $\mathbb{P}^{n_s+n_e-1}$, the inequalities carve out a *cosmological polytope* $\mathcal{P}_{\mathcal{G}}$ associated to the graph $\mathcal{G}$, with the equalities determining its facets, *i.e.* its codimension-1 boundaries [55]. The highest codimension boundaries that these facets form intersecting each other – *i.e.* the vertices of $\mathcal{P}_{\mathcal{G}}$ – can be written as

$$\left\{\mathbf{x}_{s_e} - \mathbf{y}_e + \mathbf{x}_{s'_e}, \mathbf{x}_{s_e} + \mathbf{y}_e - \mathbf{x}_{s'_e}, -\mathbf{x}_{s_e} + \mathbf{y}_e + \mathbf{x}_{s'_e}, \right\}_{e \in \mathcal{E}}, \tag{2}$$

where $\{\mathbf{x}_s, s \in \mathcal{V}\} \cup \{\mathbf{y}_e, e \in \mathcal{E}\}$ are vectors which constitute the canonical basis for $\mathbb{R}^{n_s+n_e}$.

A cosmological polytope $\mathcal{P}_{\mathcal{G}}$ is characterised by a unique, up to an overall normalisation, differential form, the *canonical form*, with a rational coefficient. The rational coefficient, named *canonical function*, is such that the denominator is given by the product of all the linear polynomials $\{q_{\mathfrak{g}}(\mathcal{Y}), \mathfrak{g} \subseteq \mathcal{G}\}$, while the numerator $\mathfrak{n}_\delta$ is a polynomial of degree $\delta$ that identifies the locus of the intersections of the hyperplanes $\{q_{\mathfrak{g}}(\mathcal{Y}) = 0, \mathfrak{g} \subseteq \mathcal{G}\}$ *outside* of $\mathcal{P}_{\mathcal{G}}$:

$$\begin{aligned} \omega\left(\mathcal{Y}, \mathcal{P}_{\mathcal{G}}\right) &= \Omega\left(\mathcal{Y}, \mathcal{P}_{\mathcal{G}}\right) \langle \mathcal{Y} d^{N-1} \mathcal{Y} \rangle \\ &= \frac{\mathfrak{n}_\delta(\mathcal{Y})}{\prod_{\mathfrak{g} \subseteq \mathcal{G}} q_{\mathfrak{g}}(\mathcal{Y})} \langle \mathcal{Y} d^{N-1} \mathcal{Y} \rangle, \end{aligned} \tag{3}$$

where: $N := n_s + n_e$; $\delta$ is the degree of the polynomial $\mathfrak{n}_\delta$ which is fixed to be $\delta = \tilde{\nu} - n_s - n_e$ – with $\tilde{\nu}$ being the number of facets of $\mathcal{P}_{\mathcal{G}}$ or, equivalently, the degree of the denominator – by projectivity as the canonical form has to be projective invariant; and $\langle \mathcal{Y} d^{N-1} \mathcal{Y} \rangle$ is the measure in $\mathbb{P}^{N-1}$. The facets, *i.e.* the linear denominators, $\{q_{\mathfrak{g}}(\mathcal{Y}), \mathfrak{g} \subseteq \mathcal{G}\}$ are identified by the covectors $\{\mathcal{W}^{(\mathfrak{g})}, \mathfrak{g} \subseteq \mathcal{G}\}$, which are such that $q_{\mathfrak{g}}(\mathcal{Y}) = \mathcal{Y}^I \mathcal{W}_I^{(\mathfrak{g})}$. The numerator $\mathfrak{n}_\delta$ is fixed by the compatibility conditions among the facets [61].

Importantly, the canonical function $\Omega\left(\mathcal{Y}, \mathcal{P}_{\mathcal{G}}\right)$ coincides with the flat-space wavefunction contribution associated to the graph $\mathcal{G}$ for a conformally-coupled scalar with polynomial interactions [55]. It can be mapped into the wavefunction contribution in an arbitrary FRW cosmology via the integral map [55]

$$\widetilde{\Omega}\left(\mathfrak{X}, \mathcal{P}_{\mathcal{G}}\right) = \prod_{s \in \mathcal{V}} \left[ \int_{X_s}^{+\infty} dx_s \tilde{\lambda}(x_s - X_s) \right] \int_\Gamma \prod_{e \in \mathcal{E}^{(L)}} [dy_e\, y_e] \mu_d(y_e, P_s) \Omega\left(\mathcal{Y}, \mathcal{P}_{\mathcal{G}}\right), \tag{4}$$

where: $\mathfrak{X} := (\{(X_s, P_s)\}_{s \in \mathcal{V}}, \{y_e\}_{e \in \mathcal{E} \setminus \mathcal{E}^{(L)}})$ parametrises the kinematic space, with $X_s := \sum_{j \in s} |\vec{p}^{(j)}|$, $P_s := |\sum_{j \in s} \vec{p}^{(j)}|$ for all the sites in a loop substructure of $\mathcal{G}$,[6] $y_e = |\sum_{j \in s_e} \vec{p}^{(j)}|$ if $e$ is an edge in a tree substructure; the measure $\tilde{\lambda}(x_s - X_s)$ encodes the details of the cosmology – for a power law cosmology, $\tilde{\lambda} \sim (x_s - X_s)^{\alpha_s - 1}$; $\mathcal{E}^{(L)} \subseteq \mathcal{E}$ is the subset of edges in the loops of $\mathcal{G}$; $\mu_d(y, \mathcal{X})$ is the integration measure over $y$ which is always positive inside the integration domain $\Gamma$ and vanishes on its boundary; the contour of integration $\Gamma$ is given by the region identified by the positivity of the volume of the simplex in $\mathbb{R}^{n_e^{(L)}}$ with the length of its edges given by $\{y_e, e \in \mathcal{E}^{(L)}\}$ and the rotational invariants $\{P_s, s \in \mathcal{V}\}$, as well as by the positivity of the volumes of all its faces at all codimensions [54].

A comment is in order. The representation (4), as described so far, holds for conformally-coupled scalars with polynomial interactions. It can be generalised to the wavefunction for certain light states [56] as well as to the actual correlators [53]. For the details of these

---

[6]Note that if just one external momentum is incident on $s$, then $P_s = X_s$.

constructions, we refer to the original papers. What is worth to point out is that it is possible to write a single integral formula for all these cases. In particular, for power law cosmologies[7]

$$\mathcal{I}_{\mathcal{G}}[\alpha,\beta,\tau;\mathfrak{X}] = \prod_{s\in\mathcal{V}}\left[\frac{i^{\alpha_s}}{\Gamma(\alpha_s)}\right]\int_{\mathbb{R}^{n_s}_+}\prod_{s\in\mathcal{V}}\left[\frac{dx_s}{x_s}x_s^{\alpha_s}\right]\int_{\Gamma}\prod_{e\in\mathcal{E}^{(L)}}\left[\frac{dy_e}{y_e}y_e^{\beta_e}\right]\mu_d(y_e;P)\frac{\mathfrak{n}_\delta(z,\mathcal{X})}{\prod_{\mathfrak{g}\subseteq\mathcal{G}}\left[q_\mathfrak{g}(z,\mathcal{X})\right]^{\tau_\mathfrak{g}}}, \quad (5)$$

where $z = (x,y)$ represents all the integration variables $x := (x_s)_{s\in\mathcal{V}}$ and $y := (y_e)_{e\in\mathcal{E}^{(L)}}$; the sets of parameters $\alpha := (\alpha_s)_{s\in\mathcal{V}}$ and $\beta := (\beta_e)_{e\in\mathcal{E}^{(L)}}$ depend on the cosmology and on the states involved in the process, and can be analytically continued for convergence. Note that, with respect to (4), the integration over the site-weights has been shifted by the rotational invariants $\{X_s, s\in\mathcal{V}\}$, and hence the linear polynomials $q_\mathfrak{g}(z,\mathcal{X})$ are no longer homogeneous in $\mathcal{Y}$:

$$q_\mathfrak{g}(z,\mathcal{X}) := \sum_{s\in\mathcal{V}_\mathfrak{g}}x_s + \sum_{e\in\mathcal{E}^{ext}_{\mathfrak{g},L}}y_e + \mathcal{X}_\mathfrak{g},$$

$$\mathcal{X}_\mathfrak{g} := \sum_{s\in\mathcal{V}_\mathfrak{g}}X_s + \sum_{e\in\mathcal{E}^{ext}_{\mathfrak{g},0}}y_e. \quad (6)$$

Finally, the parameters $\{\tau_\mathfrak{g}, \mathfrak{g}\subseteq\mathcal{G}\}$ typically acquire integer values in the physical cases but, together with the other parameters $(\alpha,\beta)$, they can be analytically continued to make the integral $\mathcal{I}_\mathcal{G}$ in (5) convergent; the numerator $\mathfrak{n}_\delta$ is a polynomial in $z$ of degree $\delta := \tau - n_s - n_e$, with $\tau = \sum_{\mathfrak{g}\subseteq\mathcal{G}}\tau_\mathfrak{g}$ and it is fixed via the compatibility conditions among the subgraphs [61].

Importantly the physical kinematics is determined by the strict positivity of all the elements of $\mathcal{X} := \{\mathcal{X}_\mathfrak{g}, \mathfrak{g}\subseteq\mathcal{G}\}$: For such kinematics, the contour of integration does not intersect any of the loci where the denominator vanishes, and hence the integral can show singularities just for small or large values of the integration variables. Were zero or negative values be allowed, then the contour of integration intersects the singularities of the integrand, with the $i\epsilon$-prescription[8] coming to the rescue by making the integral well-defined.[9] The integration measure $\mu_d(y_e;P)$ also can contribute just in the infrared and the ultraviolet given that, as specified earlier, it is positive inside the region of integration and vanishes at its boundary whose cusps represent the infrared regions – see [54]. For the time being, we will consider the physical kinematics unless otherwise specified.

## 3 The IR behaviour of cosmological integrals

The asymptotic behaviour of a given integral in the class (5) is captured by a *nestohedron* [54], a polytope defined in $\mathbb{P}^{n_s+n_e}$ as the Minkowski sum of the simplices resulting from the convex hulls of vertices given by the powers of each linear polynomial in the denominator, all of which turns out to be simplices in different codimensions – see [71, 80] as mathematical references on the nestohedra. A realisation of this nestohedron is as the sequential truncation of the top-dimensional simplex via lower-dimensional simplices corresponding to subset of its faces of all codimensions, with each facet corresponding to tubing of the underlying graph [54] – see Figure 2.

---

[7]For more general conformally flat backgrounds, the expression (5) holds modulo the substitution of the measure for the site-weight integration with a suitable function.

[8]For an extensive discussion about the $i\epsilon$-prescription in cosmology, see [79].

[9]This is similar to what happens for flat-space Feynman integrals: In the physical region, they need the $i\epsilon$-prescription for being well-defined, or to be analytically continued to Euclidean momenta. For the cosmological integrals (5), they are already defined for Euclidean momenta, as they are related to observables on a space-like surface. The analytic continuation referred in the text, is equivalent to a change of signature on such a surface.

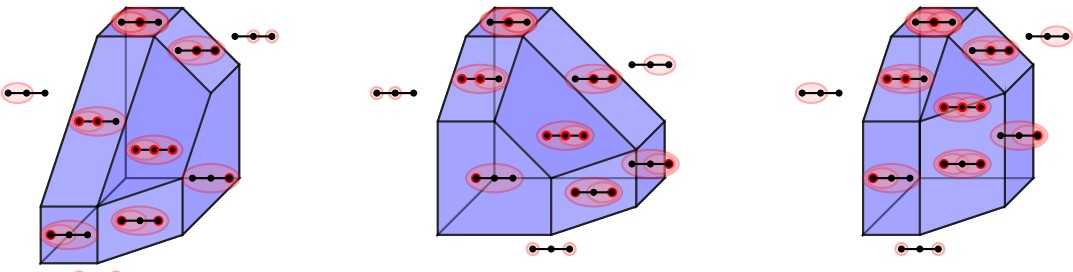

Figure 2: Example of nestrohedra. It encodes the asymptotic structure of the three-site line graph. Starting with the top-dimensional simplex, whose facets are associated to tubings, it can be realised as a sequential truncation of such a simplex based on the underlying subgraph. The faces of the nestohedron are associated then to overlapping tubings. This construction is implemented to each of the terms in which the integrand can be triangulated (*on the left and center*). The full convergence region is given by the overlap between the nestohedra associated to the chosen triangulation (*on the right*) [54].

Strictly speaking, such a nestohedron describes the asymptotic behaviour of an integral whose numerator is a polynomial of degree 0. However, note that the Mellin measure has associated a point being just a monomial: Its Minkowski sum with the nestohedron just translates the nestohedron itself. Hence, for non-trivial numerators, one can consider them as sum of monomials: Each of them would shift differently the nestohedron and, thus, the convergence region is given by the overlap among all these shifted copies of the nestohedron [54, 62]. Alternatively, it is possible to use a canonical form triangulation which does not introduce any spurious singularity [61], consider the nestohedron associated the individual simplex shifted by the point associated to the Mellin measure: The convergence region (and consequently the divergent directions) are then given by the overlap of all these shifted nestohedra corresponding to each simplex [54].

As the nestohedra determine the convergence region of the integral, their facets are associated with the divergent directions. Hence, their facet structure, together with their compatibility conditions, provide full knowledge of the asymptotic behaviour of a given cosmological integral: As we will review shortly, the co-vectors that identifies the facets of a nestohedron, allow isolating the divergences into sectors and, then, focus only on those sectors contributing to the divergence of interest. A sector encoding the divergence along a direction given by a facet, is identified by all the facets which are compatible with it.

As a further remark, note that in the case of tree graphs, these integrals are purely Mellin transforms of a rational function and the asymptotic analysis works as just described. For loop graphs, the cosmological integrals have additional integration in a region determined by the non-negativity condition of the volume of certain simplices and all their faces. Hence, the relevant divergent directions emerge from the intersection of the nestohedron analysis with this region.

**Tree level** Given a tree graph $\mathcal{G}$, the facets of the associated nestohedron are identified by the set of covectors $\left\{ \mathfrak{W}^{\prime(j_1)}, \mathfrak{W}^{(j_1 \cdots j_{n_s^{(\mathfrak{g})}})}, j_k \in [n_s - n_s^{(\mathfrak{g})} + k], k \in [1, n_s^{(\mathfrak{g})}] \right\}$ [54]:

$$\mathfrak{W}^{\prime(j_1)} = \begin{pmatrix} \lambda^{\prime(j_1)} \\ -\mathfrak{e}_{j_1} \end{pmatrix}, \qquad \mathfrak{W}^{(j_1 \cdots j_{n_s^{(\mathfrak{g})}})} = \begin{pmatrix} \lambda^{(j_1 \cdots j_{n_s^{(\mathfrak{g})}})} \\ \mathfrak{e}_{j_1 \cdots j_{n_s^{(\mathfrak{g})}}} \end{pmatrix}, \tag{7}$$

where $\mathfrak{e}_{j_1 \cdots j_{n_s^{(\mathfrak{g})}}}$ are vectors in $\mathbb{R}^{n_s}$ such that their $j_k$ entries, corresponding to the num-

ber of site tubings, are 1 while the other entries are 0, while $\lambda'^{(j_s)} := -\alpha_s^{(R)}$ and $\lambda^{(j_1 \cdots j_{n_s^{(\mathfrak{g})}}(\mathfrak{g}))} = \sum_{s \in \mathcal{V}_\mathfrak{g}} \alpha_s^{(R)} - n^{(j_1 \cdots j_{n_s^{(\mathfrak{g})}}(\mathfrak{g}))} - \alpha_s^{(R)}$ and $n^{(j_1 \cdots j_{n_s^{(\mathfrak{g})}}(\mathfrak{g}))}$ being respectively the real part of the Mellin parameter $\alpha_s$ and the number of overlapping tubings associated to the facet identified by $\mathfrak{W}^{(j_1 \cdots j_{n_s^{(\mathfrak{g})}}(\mathfrak{g}))}$.

A covector (7) identifies a divergent direction if the related $\lambda$ is non-negative: When it vanishes, it identifies a logarithmic divergence, while when it is positive the divergence is of power-law type. Importantly, there is a strict hierarchy among the $\lambda$'s, with the ones with higher $n_s^{(\mathfrak{g})}$ greater than the ones with lower. This implies that if one of the lambda's is zero, identifying a logarithmic divergence, all the other with smaller $n_s^{(\mathfrak{g})}$ are negative and hence they do not contribute to this divergence.

In this paper we will consider the integral decomposed according to the canonical form triangulations in [61]

$$\mathcal{I}_\mathcal{G} = \sum_{\{\mathfrak{G}_c\}} \mathcal{I}_{\mathfrak{G}_\circ}[\alpha; \mathfrak{G}_c], \quad \mathcal{I}_{\mathfrak{G}_\circ}[\alpha; \mathfrak{G}_c] = \int_0^{+\infty} \left[ \frac{dx}{x} x^\alpha \right] \left[ \prod_{\mathfrak{g} \in \mathfrak{G}_\circ} \frac{1}{q_\mathfrak{g}(x, \mathcal{X})} \right] \left[ \prod_{\mathfrak{g}' \in \mathfrak{G}_c} \frac{1}{q_{\mathfrak{g}'}(x, \mathcal{X})} \right], \quad (8)$$

where $[dx\, x^{\alpha-1}]$ is a short notation for the integration measure on the site weights, $\mathfrak{G}_\circ$ is a chosen set of subgraphs such that their sequential residues of the integrand is zero *irrespectively of the order* in which they are taken; while $\mathfrak{G}_c$ is a set of $n_s + n_e - k$ subgraphs that are not contained in $\mathfrak{G}_\circ$ and define a $(n_s + n_e - k)$-codimension face of $\mathcal{P}_\mathcal{G}$, with $\{\mathfrak{G}_c\}$ being all possible of such sets – they identify a subset of compatible sequential residues.

As mentioned earlier, the set of compatible facets identifies the sectors in which the integration region can be divided to isolate the divergences. Let $\Delta_\mathfrak{W}$ be the sector identified by the set $\mathfrak{W}_c$ of compatible co-vectors. Then the integral $\mathcal{I}_{\mathfrak{G}_\circ}[\alpha, \mathfrak{G}_c]$ in (8) can be parametrised in the sector $\Delta_\mathfrak{W}$ as

$$\mathcal{I}_{\Delta_{\mathfrak{W}_c}} = \int_0^1 \prod_{\mathfrak{W} \in \mathfrak{W}_c} \left[ \frac{d\zeta_\mathfrak{W}}{\zeta_\mathfrak{W}} \zeta_\mathfrak{W}^{-\lambda_\mathfrak{W}} \right] \prod_{\mathfrak{g} \in \mathfrak{G}_\circ} \prod_{\mathfrak{g}' \in \mathfrak{G}_c} \frac{1}{\left[ q_\mathfrak{g}(\zeta, \mathcal{X}) \right]^{\tau_\mathfrak{g}} \left[ q_{\mathfrak{g}'}(\zeta, \mathcal{X}) \right]^{\tau_{\mathfrak{g}'}}}, \quad (9)$$

where the variables $\{\zeta_\mathfrak{W}, \mathfrak{W} \in \mathfrak{W}_c\}$ are related to the original ones $\{x_s\, s \in \mathcal{V}\}$ via

$$x_s \longrightarrow \prod_{\mathfrak{W} \in \mathfrak{W}_c} \zeta_\mathfrak{W}^{-\mathfrak{e}_s \cdot \mathfrak{e}_\mathfrak{W}} \quad (10)$$

– for notational convenience, here we are denoting a covector as $\mathfrak{W} := (\lambda_\mathfrak{W}, \mathfrak{e}_\mathfrak{W})^\mathsf{T}$. The sectors contributing to the infrared divergences are identified by the subsets of compatible facets in

$$\left\{ \mathfrak{W}^{(j_1 \cdots j_{n_s^{(\mathfrak{g})}}(\mathfrak{g}))}, j_k \in [j, n_s - n_s^{(\mathfrak{g})} + k], k \in [1, n_s^{(\mathfrak{g})}] \right\} \quad (11)$$

– which of them contributes depend on which $\lambda$'s are non-negative.

For logarithmic divergences, $\lambda^{(1 \cdots n_s)} = 0$ while all the other $\lambda$'s are negative. Hence, the infrared divergence is encoded in all the sectors defined via $\mathfrak{W}^{(1 \cdots n_s)}$. In this case, the divergent part coming from the sum of all the sectors containing $\mathfrak{W}^{(1 \cdots n_s)}$ can be written in terms of the projection of the integrand onto the hypersurface determined by the vanishing of all the kinematic invariants

$$\mathcal{I}_\mathcal{G}^{(0)}\Big|_{\text{div}} \sim \frac{1}{-\lambda^{(1 \cdots n_s)}} \int_{\mathbb{R}_+^{n_s}} \prod_{s \in \mathcal{V}} \left[ \frac{dx_s}{x_s} x_s^{\alpha_s - \tau_{\mathfrak{g}_s}} \right] \frac{1}{\text{Vol}\{GL(1)\}} \frac{\mathfrak{n}_\delta(x, \mathcal{X} = 0)}{\prod_{\mathfrak{g} \subseteq \mathcal{G} \setminus \{\mathfrak{g}_s\}} \left[ q_\mathfrak{g}(x, \mathcal{X} = 0) \right]^{\tau_\mathfrak{g}}}, \quad (12)$$

where $\sim$ indicates the omission of coefficients which are irrelevant for the present discussion, and we used the fact that the subgraphs $\{\mathfrak{g}_s, s \in \mathcal{V}\}$ containing a single site can be included into the Mellin measure as $q_{\mathfrak{g}_s}(x, \mathcal{X} = 0) = x_s$.

For power-law divergences, such sectors contribute to both leading and subleading divergences. The latter receive contributions from all the sectors involving at least one of the co-vectors (11) with the associated $\lambda$ non-negative. In this case, for each sector, the divergent contributions can be organised as poles in products of $k^{(j_1\cdots j_{n_s^{(\mathfrak{g})}})} - \lambda^{(j_1\cdots j_{n_s^{(\mathfrak{g})}})}$, with $k^{(j_1\cdots j_{n_s^{(\mathfrak{g})}})} \in \mathbb{Z}_+$ via a Mellin-Barnes representation of the integrals [54].

**Loops**  At loop level, the contour of integration is given by $\mathbb{R}^{n_s} \cup \Gamma$, where, as specified earlier, $\Gamma$ is given by the non-negative conditions on the volume of a simplex in $\mathbb{P}^{n_e}$ and of its faces at all codimensions [54]. In this case, the analysis of the asymptotic behaviour of the integral (8) can be carried out as for tree graphs as if the integration contour were $\mathbb{R}^{n_s+n_e}$, and then intersect it with $\Gamma$: The divergent directions are then given by the ones compatible with the contour $\Gamma$. It turns out that the infrared divergences in the edge-weight integration are given by certain cusps which are located when any of the edge weights vanishes individually, one for each loop [54].

# 4  Subtracting IR divergencies

This combinatorial picture allows to systematically isolate the infrared divergences in the cosmological integrals. In this section we develop a similarly systematic procedure that removes such divergences and defines infrared finite quantities.

## 4.1  Tree-level subtractions

As in general the canonical function of a polytope is a rational function with a non-trivial numerator, it is convenient to consider one of its triangulations with no spurious boundaries and study the asymptotic behaviour of the integral of each simplex following what it has been discussed in the previous section.

Each of the simplices in the canonical form triangulation are isomorphic to each other and can be mapped into each other via a local coordinate transformation [59]. This feature reflects into the structure of the associated nestohedra which can be similarly mapped into each other. This implies that the nestohedra associated to the simplices can also be mapped into each other and, consequently, they show the same divergent directions, with their co-vectors $\{\mathfrak{W}^{(\mathfrak{g})}, \mathfrak{g} \subseteq \mathcal{G}\}$ that differ just for the projective component, *i.e.* for the degree of divergence along the same directions, which is fixed by the number of tubings associated to the relevant facets. As a consequence, the polytope identified by the overlap of the nestohedra is identified by the covectors

$$\mathfrak{W}'^{(j_1)} = \begin{pmatrix} \lambda'^{(j_1)} \\ -\mathfrak{e}_{j_1} \end{pmatrix}, \qquad \mathfrak{W}^{(j_1\cdots j_{n_s^{(\mathfrak{g})}})} = \begin{pmatrix} \min\{\lambda^{(j_1\cdots j_{n_s^{\mathfrak{g}}})}\} \\ \mathfrak{e}_{j_1\cdots j_{n_s^{(\mathfrak{g})}}} \end{pmatrix}, \tag{13}$$

with the $\min\{\lambda^{(j_1\cdots j_{n_s^{(\mathfrak{g})}})}\}$ being the minimum taken among all the nestohedra for the covectors with the same $\mathfrak{e}_{j_1\cdots j_{n_s^{(\mathfrak{g})}}}$.

Let us separately focus on each integral emerging from the canonical form triangulation of the cosmological polytope, by taking the one returning the OFPT representation. Despite this choice is not necessary, it allows for a more straightforward formulation of the subtraction rules as diagrammatic operations.[10]

---

[10]Indeed, as the asymptotic analysis does not depend on the specific triangulation chosen, the same holds for the procedure that leads to the finite quantity. Also, using tubings and markings as in [61], it is possible to assign graphical rules also for the subtractions starting from other triangulations. However, as the main point of this work is to provide a proof of concept, we leave this to future work.

Let us also treat separately logarithmic and power-law divergences, as in the discussion of the infrared asymptotics.

**Logarithmic divergences**    As we saw in the previous section, the infrared logarithmic divergences are encoded in all the sectors containing the covector $\mathfrak{W}^{(1\cdots n_s)}$ which can be written all together as

$$
\begin{aligned}
\mathcal{I}_{\mathfrak{G}_\circ}^{(0)}\Big|_{\mathrm{div}}[\alpha,\mathfrak{G}_c] &= \frac{1}{-\lambda^{(1\cdots n_s)}}\int_{\mathbb{R}_+^{n_s}}\prod_{s\in\mathcal{V}}\left[\frac{dx}{x}x^{\alpha-\tau_{\mathfrak{g}_s}}\right]\frac{1}{\mathrm{Vol}\{GL(1)\}} \\
&\quad\times\frac{1}{\left[q_{\mathcal{G}}(x,\mathcal{X}_{\mathcal{G}}=0)\right]^{\tau_{\mathcal{G}}}\prod\limits_{\mathfrak{g}\in\mathfrak{G}_c\backslash\{\mathfrak{g}_s\}}\left[q_{\mathfrak{g}}(x,\mathcal{X}_{\mathfrak{g}}=0)\right]^{\tau_{\mathfrak{g}}}} \\
&= \int_{\mathbb{R}_+^{n_s}}\prod_{s\in\mathcal{V}}\left[\frac{dx}{x}x^{\alpha-\tau_{\mathfrak{g}_s}}\right]\frac{1}{\left[q_{\mathcal{G}}(x,\mathcal{X}_{\mathcal{G}}=0)\right]^{\tau_{\mathcal{G}}}\prod\limits_{\mathfrak{g}\in\mathfrak{G}_c\backslash\{\mathfrak{g}_s\}}\left[q_{\mathfrak{g}}(x,\mathcal{X}_{\mathfrak{g}}=0)\right]^{\tau_{\mathfrak{g}}}},
\end{aligned}
\tag{14}
$$

where we have restored the full integration by rewriting the pole $1/(-\lambda^{(1\cdots n_s)})$ an integral form.

The sum over $\{\mathfrak{G}_c\}$ has then the very same divergent structure of the integrals associated to the graphs with one site and one edge less obtained by collapsing two adjacent sites, assigning to the newly generated site a weight given by the sum of the weight of the sites that have been collapsed, and shifting the Mellin parameter associated to such site-weight integration by the related $-\tau_{\mathfrak{g}}$, and summing over all the pairs of adjacent sites

$$
\mathcal{I}_{\mathcal{G}}^{(0)}\Big|_{\mathrm{ct}} = \sum_{\{e\}\in\tilde{\mathcal{E}}}(-1)^h\int_{R_+^{n_s-2h}}\left[\frac{dx}{x}x^\alpha\right]\int_{R_+^{2h}}\left[\frac{dx_e}{x_e}x_e^{\alpha-\tau_e}\right]\Omega(\{x_{\bar{s}}\};\mathcal{X}),
\tag{15}
$$

where $\tilde{\mathcal{E}}$ is the set of all edges that can be contracted simultaneously, $h$ is the number of elements in $\{e\}$, and

$$
\left[\frac{dx}{x}x^\alpha\right] := \prod_{i=1}^{n_s-2h}\frac{dx_i}{x_i}x_i^\alpha,
$$

and similarly for the second integral. Above, $\bar{s}\in\{\mathcal{V}\backslash\{s_e,s_e'\}\}\cup\{s_e+s_e'\}$.

Then, an infrared finite computable can be defined as

$$
\tag{16}
$$

where the thick graph on the left-hand side is the infrared finite computable.

To summarise, the action of contracting a pair of sites in a diagram $\mathcal{G}$ is associated to a square site in the new diagram with the respective new site weight being the sum of the weights of the two sites being contracted. Then, to construct the thick graph we take the original graph, and subtract all the diagrams in which we contracted one pair adjacent sites, then we add all the diagrams in which we contracted two pairs of adjacent sites, and repeat

this process a total of $h$ times, which is when we are contracting a maximum pairs of adjacent sites.

Below we show some diagrammatic examples, which will be more detailed in the appendix. The two site diagram:

$$
\underset{x_1 \quad\quad x_2}{\rule{0pt}{0pt}} = \underset{x_1 \quad\quad x_2}{\rule{0pt}{0pt}} - \underset{x_1 + x_2}{\rule{0pt}{0pt}} \,. \tag{17}
$$

The three-site chain diagram:

$$
\underset{x_1 \quad x_2 \quad x_3}{\rule{0pt}{0pt}} = \underset{x_1 \quad x_2 \quad x_3}{\rule{0pt}{0pt}} - \underset{x_1 \ x_2 + x_3}{\rule{0pt}{0pt}} - \underset{x_1 + x_2 \ x_3}{\rule{0pt}{0pt}} \,. \tag{18}
$$

The four-site chain diagram:

$$
\underset{x_1 \quad x_2 \quad x_3 \quad x_4}{\rule{0pt}{0pt}} = \underset{x_1 \quad x_2 \quad x_3 \quad x_4}{\rule{0pt}{0pt}}
$$

$$
- \underset{x_1 + x_2 \ x_3 \quad x_4}{\rule{0pt}{0pt}} - \underset{x_1 \ x_2 + x_3 \ x_4}{\rule{0pt}{0pt}} \tag{19}
$$

$$
- \underset{x_1 \quad x_2 \ x_3 + x_4}{\rule{0pt}{0pt}} + \underset{x_1 + x_2 \ x_3 + x_4}{\rule{0pt}{0pt}} \,.
$$

**Power-law divergences**   Power-law divergences are a little more intricate. There is a larger number of sectors which contribute to the divergent behaviour of the integral, concretely the ones identified by the adjacent covectors $\{\mathfrak{W}^{(j_1\cdots j_{n_s^{(\mathfrak{g})}})}\}$ such that $\lambda^{(j_1\cdots j_{n_s^{(\mathfrak{g})}})} \geq 0$. Recall that there is a hierarchy among the $\lambda$'s such that is larger the one associated to larger $n_s^{(\mathfrak{g})}$. This implies that it is enough to identify the smaller positive $\lambda$ or which of them vanish to know all the covectors that bound the sectors containing the divergences. As mentioned above, for each sector the divergences can be extracted by representing the integrand via a Mellin-Barnes representation – for more detail see [54] as well as the two-site tree example in the appendix. This implies that, depending on the degree of divergences along the relevant directions. For integer values of (at least some of the $\lambda$'s) multiple poles in the regulator can arise. The general logic follows what discussed in the above paragraph about the logarithmic divergences: One considers the integrals with trivial numerators obtained via a canonical form triangulation of the cosmological polytope, for each of these integrals perform the asymptotic analysis via the underlying nestohedron and identifies the sectors containing the divergences, and for each sector single out the divergences via a (multiple) Mellin-Barnes representation of the integrand. The precise structure indeed changes depending on the degree of divergence and the sector contributing.

The behaviour from the sectors identified by $\{\mathfrak{W}^{(j_1\cdots j_{n_s^{(\mathfrak{g})}})}\}$, can also be captured by graphs obtained by collapsing two adjacent sites, similarly as for the logarithmic divergences – this can be understood as all such sectors contain the infinite region for all the site weights.

When sectors are identified by both $\{\mathfrak{W}^{(j_1\cdots j_{n_s^{(\mathfrak{g})}})}\}$ and $\{\mathfrak{W}'^{(j)}\}$, the related infrared divergences are captured by disconnected graphs. This can be straightforwardly understood, as these sectors contain the infinite region for just a subset of the site-weights.

The infrared finite thick graph is given by

$$
\text{(20)}
$$

where now both the collapsing and the cut operations, respectively indicated by ■ and •-+-• contains an operator whose explicit expression depends on the degree of divergences of the relevant divergent directions – see the appendix for an explicit example.

## 4.2   Loop-level subtractions

Let us now consider loop graphs. The integration now is over both the site and (loop) edge weights. As briefly reviewed earlier, the edge weight integration is over a region $\Gamma$ defined by the positivity conditions on the volume of simplices (and their faces at all codimensions) whose sides are given by the loop edge themselves as well as by the external kinematics $\{P_s\, s \in \mathcal{V}\}$[11] [54]. Let us recall that the contour of integration $\Gamma$ restricts the asymptotic regions: In the infrared, just one edge weight per loop can be taken to be soft, while in the ultraviolet, they ought to be taken to infinity simultaneously.

Let us consider our integrals just over the loop momenta, with the $\{x_s, s \in \mathcal{V}\}$ parametrising the actual external kinematics – this occurs, for example, for the conformally coupled scalars with conformal interactions. One can construct the associated nestohedron and consider the covectors compatible with the integration contour $\Gamma$. Then, the previous statement about the asymptotic structure translates in the statement that: In the ultraviolet the covector $\mathfrak{W}^{(1\cdots n_e)}$ is the only divergent direction, and the sectors defined through it are the only one contributing to such a divergence; in the infrared instead the directions $\{\mathfrak{W}'^{(j)}\}$ corresponding to the same loop ought to be taken separately.

Let us now consider just the integration over the site weights. This is relevant for example in the case the loop integration is finite both in the infrared and in the ultraviolet – an example in the cosmological context is given by the one-loop box graph contribution to the wavefunction in $d = 3$. Then the construction of the thick graphs can follow the tree-level discussion. However, a comment is in order. Both graphs contributing to the wavefunction and the ones contributing to the in-in correlators show a total energy singularity and, when it is approached upon analytic continuation outside the physical sheet, its coefficient for both quantities is the high energy limit of the flat-space scattering amplitude. When with the subtractions we define an infrared finite quantity, ideally it would be desirable that its flat-space limit returns a flat-space infrared finite quantity. It has been long known that, in the case of the flat-space one-loop box integral, subtracting it by the triangles obtained by collapsing two vertices and replacing the suppressed propagator by the Mandelstam invariant associated to that channel, returns an infrared finite quantity – see [73] and references there in. Even in

---

[11]For graphs with just one external state for each site, $P_s = X_s \; \forall s \in \mathcal{V}$.

flat-space, how to consistently construct such infrared finite quantities is not generally known at all order perturbation theory. Here, rather than plainly mimicking the procedure outlined at tree-level, as a proof of concept, we address the question about which infrared-finite cosmological computable returns an infrared-finite flat-space quantity, in the case of the one-loop graph contribution to the wavefunction.

The one-loop box integral we are interested in is given in the OFPT representation by

$$
\begin{aligned}
\text{(box diagram)} \; &= \int_0^{+\infty}\left[\frac{dx}{x}\,x^\alpha\right]\int_{\Gamma_4}[dy\,y]\,\mu_d(y,\mathcal{X}) \\
&\times \frac{1}{q_{\mathcal{G}}\prod\limits_{s\in\mathcal{V}}q_{\mathfrak{g}_s}}\left\{\frac{1}{q_{\mathfrak{g}_{[2,1]}}}\left[\frac{1}{q_{\mathfrak{g}_{[2,4]}}}\left(\frac{1}{q_{\mathfrak{g}_{23}}}+\frac{1}{q_{\mathfrak{g}_{34}}}\right)+\frac{1}{q_{\mathfrak{g}_{23}}q_{\mathfrak{g}_{41}}}\right.\\[2mm]
&\left.\left.+\frac{1}{q_{\mathfrak{g}_{[3,1]}}}\left(\frac{1}{q_{\mathfrak{g}_{34}}}+\frac{1}{q_{\mathfrak{g}_{41}}}\right)\right]+\text{cycl. perms}\right\},
\end{aligned}
\tag{21}
$$

where $q_{[j,k]} := x_j + \ldots + x_k + y_{j-1,j} + y_{k,k+1} + \mathcal{X}_{[j,k]}$ and $\mathcal{X}_{[j,k]} := X_j + \ldots + X_k$, and $\Gamma_4$ is the integration contour obtained from the positivity conditions on the volume of a simplex in $\mathbb{P}^4$ and of all its faces – see [54]. The leading divergent direction in the site weight integration is identified by $\mathfrak{W}^{(1\cdots4)} := \left(4\alpha - 8, \mathfrak{e}_{1\cdots4}\right)^{\mathsf{T}}$ for each of the simplices and hence it possesses a logarithmic singularity for $\alpha = 2$. Were we to follow the analysis from sector decomposition, as for the tree-level case discussed earlier, then the associated thick graph would be defined by subtracting all the possible way of collapsing two adjacent sites, where the collapsing operation as defined for the tree case

$$
\text{(eq. 22 diagrams)}
\tag{22}
$$

Some comments are now in order. First, note that each triangle diagram depends just on three edge weights, rather than the original four, but the integration is still over the four edge weights: The integration along this missing edge weight can be performed returning the integration measure and the integration contour of a triangle graph – see [54]. Secondly, this thick graph stays ultraviolet finite both in the site and edge weight integrations. Furthermore, as anticipated, were we to take the flat-space limit, we *would not* obtain the known flat-space infrared safe combination among box and triangle diagrams: The collapsing operation implies a shift by one of the Mellin parameter in the integration measure for the weights associated to the two sites that have been collapsed (maintaining the correct dimensionality). So this procedure returns a computable that is infrared finite at a generic point in the physical kinematic

region but develops infrared divergences due to soft and collinear limits in the edge weights in the flat-space limit.

Alternatively, we can take into account the behaviour of the integral *outside* the physical region and design a subtraction which is still finite in the flat-space limit:

$$
\begin{aligned}
&\text{(graphical equation)} \tag{23}
\end{aligned}
$$

where $s_{ij} := x_i + x_j + X_i + X_j + P_{ij}$, $P_{ij} := |\vec{p}_i + \vec{p}_j|$, the red dashed circle indicate that the singularity associated to the subgraph it identifies has been removed, and the permutations indicated with *perm.* are for the term appearing immediately before. Note that the introduction of the bubble introduces a spurious singularity in the ultraviolet. It can be removed via the introduction of further terms. It is straightforward to check that, in the flat-space limit, just the first five terms in the right-hand-side contribute, returning the flat-space infrared finite combination of Feynman graphs.

Let us close this section with a comment on (23). It represents a proof of concept that a quantity which is both infrared finite in an expanding Universe stays infrared finite in its flat-space limit. Comparing with (22), if one side it has the nice feature just mentioned that (22) does not have, on the other side it is utterly more complicated, and it would be desirable to have a simpler object with an infrared-finite flat-space limit. If such a computable exists, a similar analysis to the one discussed in this paper but for unphysical kinematics, *i.e.* allowing for some energies to be negative, would be a systematic approach to definte it. However, it would require some extra care because, for unphysical kinematics, the polynomial appearing in the integrand are no longer all sums of positive quantities and at least some loci given by the vanishing of the polynomials in the denominators, intersect the integration contour. We leave this to future work.

# 5 Conclusion & outlook

Cosmological observables are plagued by infrared divergences, especially when light states are involved. For massless scalars in de Sitter space-time, there are strong indications that these divergences resum. What is instead the fate of such divergences in general FRW cosmologies

and for more general light states, to our knowledge, it is not fully understood. In particular, it is not fully understood which feature the analytic structure of an observable needs to have for the infrared divergences to be guaranteed to resum (or, vice versa, for the theory to be pathological) as well as how to define our observables from first principle in such a way that it is infrared finite from the start.

In this paper, we continue the route traced in [54] for having a deeper understand and control over the asymptotic behaviour of the perturbative contributions to physical observables. We exploit the combinatorics behind the infrared behaviour of cosmological integrals to consistently define, graph by graph, a computable which is infrared finite via subtractions. Such subtraction can be understood in terms of operations on graphs: Given a certain graph, an associated infrared finite quantity is obtained by subtracting all the graphs obtained by collapsing two adjacent sites. As we stressed in the introduction, we do not propose this subtraction procedure to be a way of renormalising the infrared divergences, rather it can signal the existence of a novel computable whose perturbative expansion is given by our thick graphs (or it is anyhow strictly related). The existence of the thick graph rules open the question about a more generally defined computable whose originates such rules. We have left its possible independent, first principle, definition to future work. As a further remark, across this paper we have referred to it as a *computable*. This is related to two basic facts: First, there is the general difficulty in the proper definition of an observable at future infinity in an expanding background – see for example [81, 82]; secondly, and more importantly, its physical content as well as a more first principle relation to a probability distribution still needs to be explored. This is also left to future work. Finally, these studies allow for a more in-depth understanding of the analytic structure of cosmological observables and set the grounds on one side to extract physical information from perturbative infrared finite computables and asking more sharply what objects we can define that can help us to understand the early-time physics, and on the other to address the two questions we posed above.

## Acknowledgments

It is a pleasure to thank Subodh Patil and William Torres Bobadilla for valuable discussions. P.B. would also like to thank the developers of SageMath [83], Maxima [84], Polymake [85–88], TOPCOM [89], and Tikz [90]. P.B. would like to thank Dieter Lüst and Gia Dvali for making finishing it possible, as well as the Instituto Galego de Física de Altas Enerxías of the Universidade de Santiago de Compostela for hospitality during the second part of this work.

**Funding information**   P.B. and F.V. have been partially supported during the first part of this work by the European Research Council under the European Union's Horizon 2020 research and innovation programme (No 725110).

## A   Explicit examples

In this appendix, we illustrate our subtraction procedure and the perturbative computables we define through it with some explicit examples.

**The two-site tree graph**   Let us consider the two-site tree graph integral:

$$I_2^{(2)} = \int_{\mathbb{R}_+^2} \left[ \frac{dx}{x} \, x^\alpha \right] \frac{1}{(x_1 + x_2 + \mathcal{X}_{\mathcal{G}})(x_1 + \mathcal{X}_{\mathfrak{g}_1})(x_2 + \mathcal{X}_{\mathfrak{g}_2})} \, . \tag{A.1}$$

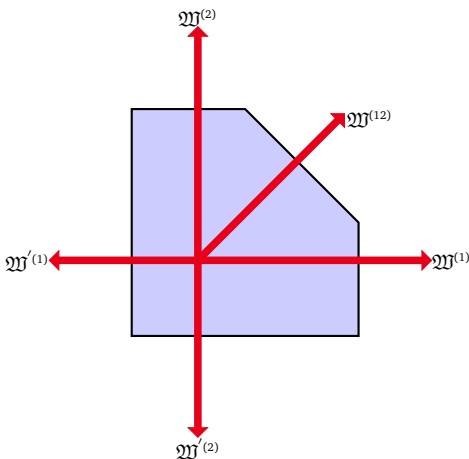

Figure 3: Nestohedron associated to the two-site tree graph. It shows divergences along the normal vectors $\{\mathfrak{W}^{(12)}, \mathfrak{W}^{(j)}, \mathfrak{W}'^{(j)}, j = 1, 2\}$ that identify its facets. The potential IR divergences are captured by the sectors containing $\mathfrak{W}^{(12)}, \mathfrak{W}^{(1)}, \mathfrak{W}^{(2)}$ [54].

It can diverge along the following directions [54] – see Figure 3:

$$\mathfrak{W}^{(12)} = \begin{pmatrix} 2\alpha - 3 \\ \mathfrak{e}_{12} \end{pmatrix}, \qquad \mathfrak{W}^{(j)} = \begin{pmatrix} \alpha - 2 \\ \mathfrak{e}_j \end{pmatrix}, \qquad \mathfrak{W}'^{(j)} = \begin{pmatrix} -\alpha \\ -\mathfrak{e}_j \end{pmatrix}, \tag{A.2}$$

with $j = 1, 2$. The integral is well-defined for values of $\alpha$ that identify a point inside the nestohedron, which implies that all the $\lambda$'s ought to be strictly negative, *i.e.* for $\alpha \in ]0, 3/2[$. It shows a logarithmic divergence in the IR for $\lambda^{(12)} := 2\alpha - 3 = 0$, which is captured by the two sectors $\Delta_{2,12}$ and $\Delta_{12,1}$, respectively delimited by the pairs $(\mathfrak{W}^{(2)}, \mathfrak{W}^{(12)})$ and $(\mathfrak{W}^{(12)}, \mathfrak{W}^{(1)})$.

Then, an IR-finite quantity can be defined by subtracting the graph obtained by collapsing the two sites onto each other, assigning to the newly generated site the weight given by the sum of the weights of the two sites that have been collapsed

$$\underset{x_1 \qquad x_2}{\bullet\!=\!=\!\bullet} = \underset{x_1 \qquad\qquad x_2}{\bullet\!-\!-\!\bullet} - \underset{x_1 + x_2}{\blacksquare} . \tag{A.3}$$

The collapsing operation implies a shift by one of the Mellin parameters in the integration measure, such that

$$\underset{x_1 + x_2}{\blacksquare} = \int_{\mathbb{R}_+^2} \left[ \frac{dx}{x} x^{\alpha-1} \right] \frac{1}{x_1 + x_2 + \mathcal{X}_{\mathcal{G}}} . \tag{A.4}$$

Note that this rule provides a subtraction scheme at integrand level. Then, the double line two-site tree graph corresponds to the integral

$$\underset{x_1 \qquad x_2}{\bullet\!=\!=\!\bullet} = -\int_{\mathbb{R}_+^2} \left[ \frac{dx}{x} x^{\alpha-1} \right] \frac{\mathcal{X}_{\mathfrak{g}_2} x_1 + \mathcal{X}_{\mathfrak{g}_1} x_2 + \mathcal{X}_{\mathfrak{g}_1} \mathcal{X}_{\mathfrak{g}_2}}{(x_1 + x_2 + \mathcal{X}_{\mathcal{G}})(x_1 + \mathcal{X}_{\mathfrak{g}_1})(x_2 + \mathcal{X}_{\mathfrak{g}_2})} . \tag{A.5}$$

As a check, we can study the asymptotic behaviour of (A.5). As discussed in [54, 62], we can partial fraction the integrand, study separately the integrals in the resulting sum, and finally obtain the region of convergence by overlapping the Newton polytopes associated to them – see Figure 4.

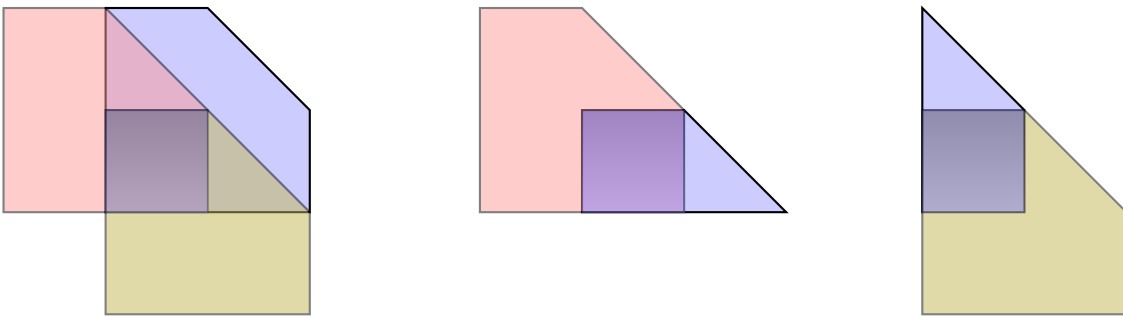

Figure 4: Convergence polytope associated to the two-site tree thick graph. It is given the overlap of the Newton polytopes of the integrals in which it can be decomposed as a sum, with in this case is a square – it is indicated by the shaded blue region. *Left:* The integral is decomposed according to the monomials in the numerator. *Center:* The numerator is decomposed as $\mathcal{X}_{\mathfrak{g}_2} x_1 + \mathcal{X}_{\mathfrak{g}_1}(x_2 + \mathcal{X}_{\mathfrak{g}_2})$. *Right:* The numerator is decomposted as $\mathcal{X}_{\mathfrak{g}_1} x_2 + \mathcal{X}_{\mathfrak{g}_2}(x_1 + \mathcal{X}_{\mathfrak{g}_1})$.

Irrespectively of the partial fraction chosen, the convergence region is given by a square with vertices $\{(1, 1-\alpha, 1-\alpha), (1, 1-\alpha, 2-\alpha), (1, 2-\alpha, 2-\alpha), (1, 2-\alpha, 1-\alpha)\}$, whose facets are identified by the co-vectors

$$\mathfrak{W}_c^{(j)} = \begin{pmatrix} \alpha - 2 \\ \mathfrak{e}_j \end{pmatrix}, \qquad \mathfrak{W}_c'^{(j)} = \begin{pmatrix} 1 - \alpha \\ -\mathfrak{e}_j \end{pmatrix}, \tag{A.6}$$

where $j = 1, 2$. The double-line two-site tree graph thus converges for $\alpha \in {]}1, 2{[}$. The point for $\alpha = 3/2$, which in the original two-site graph generated a logarithmic divergence, now lies inside the Newton polytope of the double-line two-site graph. Note that for $\alpha = 3/2$, (A.5) stays UV finite.

Finally, note that for $\alpha \geq 2$ the double-line two-site tree graph graph diverges in the infrared: This corresponds to a power-law divergence in the original two-site tree graph, to which the leading divergence has been subtracted. This is manifest in the Newton polytope picture (Fig. 4): It is a square with the same divergent directions as the pentagon associated to the original graph, except for $\mathfrak{e}_{12}$.

Let us now consider the case of a power-law divergence. For definiteness let us consider the case $\alpha = 2$, *i.e.* $\lambda^{(12)} = 1$ and $\lambda^{(j)} = 0$. The infrared divergences are now encoded into four sectors, namely $\{\Delta_{2,12}, \Delta_{12,1}, \Delta_{1',2}, \Delta_{1,2'}\}$ – *i.e.* those respectively identified by the pairs of covectors $\{(\mathfrak{W}^{(2)}, \mathfrak{W}^{(12)}), (\mathfrak{W}^{(12)}, \mathfrak{W}^{(1)}), (\mathfrak{W}^{(1')}, \mathfrak{W}^{(2)}), (\mathfrak{W}^{(1)}, \mathfrak{W}^{(2)})\}$. In the sector $\Delta_{2,12}$, the integral acquires the form

$$\mathcal{I}_{\Delta_{2,12}} = \int_0^1 \frac{d\zeta_{12}}{\zeta_{12}} \zeta_{12}^{-\lambda^{(12)}} \int_0^1 \frac{d\zeta_2}{\zeta_2} \zeta_2^{-\lambda^{(2)}} \frac{1}{(1 + \zeta_2 + \mathcal{X}_{\mathcal{G}} \zeta_2 \zeta_{12})(1 + \mathcal{X}_{\mathfrak{g}_1} \zeta_{12})(1 + \mathcal{X}_{\mathfrak{g}_2} \zeta_2 \zeta_{12})}, \tag{A.7}$$

where $x_1 = \zeta_{12}^{-1}$ and $x_2 = \zeta_2^{-1} \zeta_{12}^{-1}$. As discussed in [54], the divergent terms can be isolated via a Mellin-Barnes representation for the integrand, yielding

$$
\begin{aligned}
\mathcal{I}_{\Delta_{2,12}} = &\int_0^1 \frac{d\zeta_{12}}{\zeta_{12}} \zeta_{12}^{-\lambda^{(12)}} \int_0^1 \frac{d\zeta_2}{\zeta_2} \frac{\zeta_2^{-\lambda^{(2)}}}{1 + \zeta_2} - \mathcal{X}_{\mathfrak{g}_1} \int_0^1 \frac{d\zeta_{12}}{\zeta_{12}} \zeta_{12}^{-\lambda^{(12)}+1} \int_0^1 \frac{d\zeta_2}{\zeta_2} \frac{\zeta_2^{-\lambda^{(2)}}}{1 + \zeta_2} \\
&- \mathcal{X}_{\mathfrak{g}_2} \int_0^1 \frac{d\zeta_{12}}{\zeta_{12}} \zeta_{12}^{-\lambda^{(12)}+1} \int_0^1 \frac{d\zeta_2}{\zeta_2} \frac{\zeta_2^{-\lambda^{(2)}+1}}{1 + \zeta_2} - \mathcal{X}_{\mathcal{G}} \int_0^1 \frac{d\zeta_{12}}{\zeta_{12}} \zeta_{12}^{-\lambda^{(12)}+1} \int_0^1 \frac{d\zeta_2}{\zeta_2} \frac{\zeta_2^{-\lambda^{(2)}+1}}{(1 + \zeta_2)^2} + \dots
\end{aligned}
\tag{A.8}
$$

Repeating this analysis for all the sectors, gathering all the terms together, and restoring the original variables, it is easy to see that this divergent structure is captured by

$$\mathcal{I}_2^{(0)}\big|_{\text{ct}} = \int_0^\infty \prod_{j=1}^2 \left[\frac{dx_j}{x_j} x_j^{\alpha-1}\right]\left(1 + \mathcal{X}_{\mathcal{G}}\frac{\partial}{\partial \mathcal{X}_{\mathcal{G}}} - \frac{\mathcal{X}_{\mathfrak{g}_1}}{x_1} - \frac{\mathcal{X}_{\mathfrak{g}_2}}{x_2}\right)\frac{1}{x_1+x_2+\mathcal{X}_{\mathcal{G}}}$$
$$+ \int_0^{+\infty} \prod_{j=1}^2 \left[\frac{dx_j}{x_j} x_j^{\alpha-2}\right]\frac{x_1\mathcal{X}_{\mathfrak{g}_2}^2 + x_2\mathcal{X}_{\mathfrak{g}_1}^2}{(x_1+\mathcal{X}_{\mathfrak{g}_1})(x_2+\mathcal{X}_{\mathfrak{g}_2})}, \tag{A.9}$$

and the infrared finite thick graph can be defined as

$$\underset{x_1 \qquad x_2}{\rule{0pt}{0pt}} = \underset{x_1 \qquad x_2}{\rule{0pt}{0pt}} - \underset{x_1+x_2}{\rule{0pt}{0pt}} - \underset{x_1 \qquad x_2}{\rule{0pt}{0pt}} , \tag{A.10}$$

where now the collapsing operation includes the operator in the first term in (A.9), while the second term in (A.9) can be represented via a cut operation.[12] As stressed in the main text, while these rules are general – the thick graph can be defined in terms of collapsing and cuts – the explicit form relies on the degree of divergence (as well as on whether (some of) the $\lambda$'s are integer or not).

**Three-site tree graph**    In this case, the integrand has a non-trivial numerator. Considering its OFPT representation, the convergence of its Mellin transform is given by the overlap between the two nestohedra associated to each of the OFPT terms – see Figure 2. Following the procedure outlined above, for logarithmic divergencies, one can obtain a finite term by subtracting from the original graph the sum of the graphs obtained by collapsing all the pairs of sites connected by an edge, assigning to the newly generated site the sum of the weights of the sites that have been collapsed and shifting the Mellin parameter by $-1$ in the site weight integration related to the sites that have been collapsed:

$$\underset{x_1 \quad x_2 \quad x_3}{\rule{0pt}{0pt}} = \underset{x_1 \quad x_2 \quad x_3}{\rule{0pt}{0pt}} - \underset{x_1 \quad x_2+x_3}{\rule{0pt}{0pt}} - \underset{x_1+x_2 \quad x_3}{\rule{0pt}{0pt}} , \tag{A.11}$$

where

$$\underset{x_1 \quad x_2+x_3}{\rule{0pt}{0pt}} = \int_{\mathbb{R}_+}\frac{dx_1}{x_1}x_1^{\alpha}\int_{\mathbb{R}_+^2}\left[\frac{dx}{x}x^{\alpha-1}\right]\frac{1}{(x_1+x_2+x_3+\mathcal{X}_{\mathcal{G}})(x_1+\mathcal{X}_{\mathfrak{g}_1})(x_2+x_3+\mathcal{X}_{\mathfrak{g}_{23}})}, \tag{A.12}$$

and similarly for the third term in the right-hand-side of (A.11).

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
