# Peer review of "Cosmological Infrared Subtractions & Infrared-Safe Computables"

_SciPost Physics, doi:SciPost Phys. 18, 176 (2025)_

## Round 2 · Referee Report · Anonymous (Referee 1) · 2024-12-18

Strengths

Main Strengths:
\begin{itemize}
\item The manuscript offers a systematic, diagrammatic methodology for addressing infrared divergences in scalar cosmological observables. The use of cosmological polytopes and nestohedra to formalize divergence subtraction represents an innovative intersection of combinatorial mathematics and cosmological physics.

\item The theoretical foundation is presented systematically. The introduction effectively contextualizes the problem of infrared divergences, grounding the proposed solution in prior work on analytic structures in cosmological observables.

\item The derivations and formalism appear thorough, with explicit mathematical details provided for the subtraction rules and their application to logarithmic and power-law divergences.

\item The framework has the potential to address divergences in various perturbative settings beyond the specific examples considered in the paper. This adaptability makes it a promising tool for the broader cosmology community.

\end{itemize}

Weaknesses

Weaknesses:
\begin{itemize}
\item While the mathematical framework is robust, the physical implications of the subtraction procedure and the resulting infrared-safe quantities are not thoroughly explored. For instance, how these quantities relate to observable features of the early universe remains unclear.
\item The manuscript would benefit from a more detailed comparison with alternative methods for handling infrared divergences, such as resummation techniques or holographic renormalization approaches.
\item While the procedure is well-defined mathematically, practical implementation details (e.g., computational complexity, numerical stability, and scalability) are not discussed. These aspects are important for the adoption of the method in real-world computations.
\end{itemize}

Report

The manuscript introduces a novel procedure to address infrared divergences in scalar cosmological integrals within perturbative frameworks. By leveraging the combinatorial structures of cosmological polytopes and nestohedra, the authors propose systematic subtraction rules to define infrared-safe computable quantities. These subtraction rules rely on diagrammatic operations on weighted reduced graphs derived from Feynman diagrams. The approach is illustrated with examples, including tree and loop-level computations, to validate the effectiveness of the proposed methodology.

The manuscript meets SciPost’s criteria for significant, original contributions to theoretical cosmology. The proposed methodology is innovative, and the rigorous mathematical framework aligns well with the journal’s standards. The suggested improvements below would enhance the paper’s clarity and broader applicability, but even in its current form, the work represents a valuable contribution to the field.

I recommend the manuscript for publication in SciPost Physics, subject to the authors addressing the points raised above and below.

Requested changes

Additional comments:
\item I had to read until page 10 to understand what the authors are actually proposing to do with these ``infrared safe computables''. The intro is not clear about what the authors think one should do with these ``subtractions'', and what the meaning could possibly be of these ad-hoc constructed infrared computables. I think this is explained in the conclusions in the paragraph ``In this paper, \dots''. This discussion would be much more useful early on, maybe in the intro.
\item Another question that is left unanswered is how many other infrared computable are there and why those defined in this paper are distinct from the infinitely many I could imagine defining. Isn't is completely arbitrary what I subtract, as long as I remove the IR divergences? What makes one subtraction ``better'' than another if none of them has a clear physical meaning?
\item is it true that all the technical results apply only to a conformally coupled scalar or do the authors have some more general precise result (as opposed to an expectation)? In this regard I was a bit confused by ``Importantly, the canonical function Ω(Y,PG) coincides with the flat-space wavefunction contribution associated to the graph G for a conformally-coupled scalar with polynomial interactions'', where a conf. coupled field in Mink. just means a massless field.
\item the distinction was to clear to me between general results valid for an arbitrary diagram and specific results for a given diagram. for example, where do eq 17 and 18 come from and why are we looking at those diagrams to begin with? Are those supposed to be the simplest non-trivial examples?
\item Similar question goes for the box diagram. Do the author claim to have spelled out a systematic procedure to remove all IR divergences for all diagrams to all loops or just for the box? I did not see/understand the general procedure, if one is presented.
\item The derivation of the subtraction procedure is rigorous but dense. Consider including a flowchart or diagram summarizing the main steps of the methodology for clarity.
\item ``posses'' $ \to$``possesses''?

Attachment

Recommendation

Publish (easily meets expectations and criteria for this Journal; among top 50%)

---

## Round 2 · Referee Report · Anonymous (Referee 2) · 2024-12-30

Strengths

1- Analyzes in detail the IR safety of the wavefunction of the universe, identifying when certain wavefunction coefficients are IR divergent
2- Suggests an IR-safe observable which is finite

Weaknesses

1- Worked examples only in the appendix, and even then, origin of the IR divergence (and its subtraction) not explained physically

Report

This is a very nice paper which should be published in SciPost. The authors provide the first steps in a systematic classification of large IR effects in the wavefunction of the universe. Though they do mention the potential appearance of other IR divergences in the actual in-in observables, they focus on wavefunction coefficients, for which some of these IR divergences are absent.

What I think would greatly improve the presentation of the paper is to illustrate the abstract discussion with worked examples in the main body of the paper. The current presentation is not so nice, where the actual examples are worked out in an appendix. Moreover, some more physical insight of why the 2-site graph example has IR divergences for some range of \alpha would be interesting to see. Presumably the divergence only appears if the spacetime is evolving "fast enough" in some sense.

Finally, what is the physical interpretation of the IR-safe observable? Is it truly necessary for the wavefunction coefficient? Recall that e.g. even the 2-pt wavefunction coefficient of a massless scalar in dS has a (power-law) IR divergence, but the in-in correlator is insensitive to that. Another example is the 3-pt function of a conformally coupled scalar at tree-level. Again, there is a (this time logarithmic) divergence in the wavefunction, but not in the correlator. It'd be interesting to explain what the subtracted wavefunction is encoding physically, as it is softening correlation functions between the fields and its momenta.

Requested changes

1- I'd like the authors to reorganize the paper so that the worked examples appear in the main body of the paper
2- If they can give more physical insight to the need for an IR-finite wavefunction, as well as the reason why some background spacetimes cause large IR effects in the explicit examples, that would greatly improve the clarity of the paper.

Recommendation

Publish (easily meets expectations and criteria for this Journal; among top 50%)

---

## Round 3 · Author Response

We thank the referees for the valuable feedback on our manuscript. Below we answer the comments and concerns of the referees.
Referee 1:
We moved the part where we discuss the flat space example of infrared safe observables from the conclusion to the introduction, we feel this should motivate better why searching for infrared safe quantities can be interesting also in cosmology.
Regarding the question of how many infrared safe quantities there are, we believe this is tied to the previous point. Precisely, despite mathematically one can write many infrared safe quantities, but practically just a handful of them would be physically meaningful. Having a systematic procedure, like ours, that on top of them mimics what happens in flat-space, and whose perturbative expansion is really sourced by an observable which has its own non-perturbative definition (the Wilson loop with Lagrangian insertion). Our procedure points towards understanding the possible existence of an analogous quantity in an expanding universe.
We explicitly discussed conformally coupled scalars only. However, for external massless scalars they directly obtained from conformally coupled from the action of differential operators acting on the external kinematics, and altering the power alpha in the measure, so our thick graphs directly apply.
Regarding the non-trivial examples in eps. 16 and 20 (in the new version), we addressed this by adding the simplest diagrammatic examples in the main body instead of the examples. We altered the discussion around eq. 15 (new version) as well, in hopes this would make the derivation of the rules less dense.
Regarding the section on loops, our procedure only works for removing the secular divergences, associated with the site integration. So in principle it works for all loops. The particular example of the Box in equation 23 (new version) is a particular case, which we did not generalize, It is interesting because in the flat space limit it returns also an infrared finite observable. The generalisation of this example to higher loops is not even settled in flat-space, despite some work like the one by Anastasiou and Sterman that we cite.
Referee 2:
Regarding the worked examples, we feel that putting them fully in the main body would break the flow of the text. Therefore, we decided to put the diagrammatic part of the examples, even adding a new one, in the main body. But we would like to insist in leaving the explicit analytic computations for the appendices
Regarding the reason behind IR divergences, we expanded the second paragraph of the introduction shortly detailing what are these divergences and their physical origin.
Referee 1:
We moved the part where we discuss the flat space example of infrared safe observables from the conclusion to the introduction, we feel this should motivate better why searching for infrared safe quantities can be interesting also in cosmology.
Regarding the question of how many infrared safe quantities there are, we believe this is tied to the previous point. Precisely, despite mathematically one can write many infrared safe quantities, but practically just a handful of them would be physically meaningful. Having a systematic procedure, like ours, that on top of them mimics what happens in flat-space, and whose perturbative expansion is really sourced by an observable which has its own non-perturbative definition (the Wilson loop with Lagrangian insertion). Our procedure points towards understanding the possible existence of an analogous quantity in an expanding universe.
We explicitly discussed conformally coupled scalars only. However, for external massless scalars they directly obtained from conformally coupled from the action of differential operators acting on the external kinematics, and altering the power alpha in the measure, so our thick graphs directly apply.
Regarding the non-trivial examples in eps. 16 and 20 (in the new version), we addressed this by adding the simplest diagrammatic examples in the main body instead of the examples. We altered the discussion around eq. 15 (new version) as well, in hopes this would make the derivation of the rules less dense.
Regarding the section on loops, our procedure only works for removing the secular divergences, associated with the site integration. So in principle it works for all loops. The particular example of the Box in equation 23 (new version) is a particular case, which we did not generalize, It is interesting because in the flat space limit it returns also an infrared finite observable. The generalisation of this example to higher loops is not even settled in flat-space, despite some work like the one by Anastasiou and Sterman that we cite.
Referee 2:
Regarding the worked examples, we feel that putting them fully in the main body would break the flow of the text. Therefore, we decided to put the diagrammatic part of the examples, even adding a new one, in the main body. But we would like to insist in leaving the explicit analytic computations for the appendices
Regarding the reason behind IR divergences, we expanded the second paragraph of the introduction shortly detailing what are these divergences and their physical origin.

---

## Editorial Decision

published